# A Multi-Model Ensemble Approach for Gold Mineral Prospectivity Mapping: A Case Study on the Beishan Region, Western China

**Kaijian Wang [1], Xinqi Zheng [1,2,*], Gongwen Wang [3], Dongya Liu [1] and Ning Cui [4]**

[1] School of Information and Engineering, China University of Geosciences Beijing, Beijing 100083, China; 3dgmp@cugb.edu.cn (K.W.); dyliu@cugb.edu.cn (D.L.)

[2] Technology Innovation Center for Territory Spatial Big-Data, MNR of China, Beijing 100083, China

[3] School of Earth Sciences and Resources, China University of Geosciences Beijing, Beijing 100083, China; gwwang@cugb.edu.cn

[4] Development and Research Center of China Geological Survey, Beijing 100037, China; cning@mail.cgs.gov.cn

* Correspondence: zhengxq@cugb.edu.cn; Tel.: +86-010-8232-2116

**Abstract:** Mineral prospectivity mapping (MPM) needs robust predictive techniques so that the target zones of mineral deposits can be accurately delineated at a specific location. Although an individual machine learning algorithm has been successfully applied, it remains a challenge because of the complicated non-linear relations between prospecting factors and deposits. Ensemble learning methods were efficiently applied for their excellent generalization, but their potential has not been fully explored in MPM. In this study, three well-known machine learning models, namely random forest (RF), support vector machine (SVM), and the maximum entropy model (MaxEnt), were fused into ensembles (i.e., RF–SVM, RF–MaxEnt, SVM–MaxEnt, RF–SVM–MaxEnt) to produce a final prediction. The paper aims to investigate the potential application of stacking ensemble learning methods (SELM) for MPM. In this study, 69 hydrothermal gold deposits were split into two parts: 70% for the training model and 30% for testing the model. Then, 11 mineral prospecting factors were selected as a spatial dataset constructed for MPM. Finally, the models' performance was assessed using the receiver operating characteristic (ROC) curves and five statistical metrics. Compared with other single methods, the SELM framework showed an improved predictive performance in the model evaluation. Therefore, this finding suggests that the SELM framework is promising and should be selected as an alternative technique for MPM.

**Keywords:** stacking ensemble learning method; random forest; support vector machine; maximum entropy model; mineral prospectivity mapping; Beishan region, China

## 1. Introduction

The demand for mineral resources has grown significantly in recent years [1], mainly due to rapid industrial development in developing countries, such as China, India, and Brazil [2]. The prediction of mineral prospectivity is a multivariable decision-making tool used to draw and rank zones that have the highest potential for mineral exploration in unexplored regions [3]. Mineral predictive modelling is a vital but challenging step for the mapping of undiscovered prospective deposits in mineral prospectivity mapping (MPM). Therefore, effective modelling techniques of mineral resource exploration are increasingly critical for contributing to sustainable economic growth on the national level. As such, mapping new mineral prospectivity has become imperative, and predictive modelling provides a scientific means for delineating the intricate spatial patterns of features that are closely related to mineralisation [4].

Over the last several decades, various data-driven modelling techniques have led to optimisation and improvements in the practical application of MPM [5–7]. Statistical methods have achieved great popularity, such as weights of evidence [8], fuzzy weights of evidence [9], and Bayesian networks [10], because these models have the advantages of lucid expression and simplicity of interpretation [11,12]. In the past decade, machine learning algorithms have been efficiently applied for MPM, such as decision trees [13], artificial neural networks [1], support vector machine (SVM) [14], logistical regression [15], maximum entropy (MaxEnt) [16], and random forest (RF) [17], etc. Previous studies have shown that machine learning techniques have better predictive performance than traditional statistical techniques [1,18]. Furthermore, machine learning algorithms have a better advantage of handling non-linear relationships between known deposits and spatial layers well [19] than previously used techniques.

Furthermore, they are potent methods for managing large numbers of evidential features and are widely used in geographic information science (GIS)-based MPM to generate reliable mapping [20]. More recently, deep learning, i.e., a new method of representation learning, has allowed multiple processing layers to learn multiple levels of representation of the input and has generated state-of-the-art results in many fields [21]. Big data analytics and a deep autoencoder network [15] and convolutional neural network [22] are used to learn and mine spatial patterns from a large number of inputs for MPM. However, it should be noted that some innovative and robust methods have been recommended for MPM, but no model proves to be superior to other methods in all situations [12].

Ensemble learning methods are proven machine learning techniques that integrate various base learners [23] to achieve more accurate predictions [24]. At present, state-of-the-art ensemble methods can be grouped into three categories: bagging, boosting, and stacking [25]. These base learners are homogeneous in bagging and boosting ensembles, while they can be homogeneous or heterogeneous in stacking ensembles. Moreover, they are constructed by sequential or parallel base learners. Unlike the other two frameworks, stacking combines several types of base learning classifiers to improve generalisation performance [24]. Heterogeneous ensemble learning methods can obtain the advantages of different models, and they are more robust than homogeneous ensemble learning.

Moreover, the difficulty in model selection for MPM can also be avoided to a certain extent, as heterogeneous stacking ensemble learning methods (SELM) can gain heterogeneity by fusing any types of base model [26]. However, few scholars have paid attention to the exploration of the stacking ensemble to integrate multiple heterogeneous types of classifiers in MPM. This research fills this gap by evaluating and validating a novel ensemble learning method for mineral potential modelling. MaxEnt and RF models had been used effectively to predict the gold prospectivity in the Hezuo–Meiwu district, west Qinling orogen, China [27] and an SVM model had been used successfully to predict copper potential mapping in Kerman region, Iran [20]. Furthermore, the Beishan region is a prepared case study area that provides rich geological data to train spatial models. Therefore, these three heterogeneous ensemble techniques (RF, SVM, and MaxEnt) were selected to improve the predictive performance in the case of the Beishan region, west China. There are three main contributions of this work: First, we aimed to investigate the potential application of ensembles of RF, SVM, and MaxEnt algorithms (i.e., RF–SVM, RF–MaxEnt, SVM–MaxEnt, and RF–SVM–MaxEnt) for predictions of mineral prospectivity in the Beishan region, west China. Second, analysis of the importance of the features' given contributions can provide specific insight into the final prediction results to some extent.

## 2. Study Area and Data Preparation

### 2.1. Study Area

The Beishan region, China, is located in the north-eastern part of the Tarim Basin, in the west-central part of the Palaeozoic Tianshan–Yinshan–Great Hinggan metallogenic belt, adjacent to the Middle Tianshan Massif. It is a typical intersection of multiple geological unit structures (Figure 1a) in the previous research work [28–31]. Proterozoic, Palaeozoic, and Mesozoic strata are the ore-controlling stratum. The Beishan region comprised a Precambrian crystalline basement and overlying sedimentary rocks of

Palaeozoic–Mesozoic age. These intrusions formed from the Precambrian to the Permian and are separated by well-developed fault-related hosts and grabens [32]. The late Palaeozoic tectonic history in the region is closely related to the subduction and subsequent trapping of the southern Tianshan oceanic plate under the Junggar and Tarim plates during the Middle Ordovician through the Silurian [33]. The Beishan region can be divided into three tectonic units, bound by the Liuyuan–Daqishan and Hongshishan–Heiyingshan faults, which were affected by multiple stages of magma intrusion during the Carboniferous and Permian. Furthermore, the structure of this study area is involved, along with large-scale strong tectonic, magmatic, and metamorphic effects, which provide heat and channels for the activation, transfer, and enrichment of metal elements, and form abundant gold (Au) resources [33]. The Beishan region, one of the most significant potential Chinese gold production zones, has had more than 200 tons of gold discovered as of yet [28,29]. There is still excellent gold potential undiscovered in this study area, due to the cover of surface vegetation and complexity of the convergent zone in this region [30].

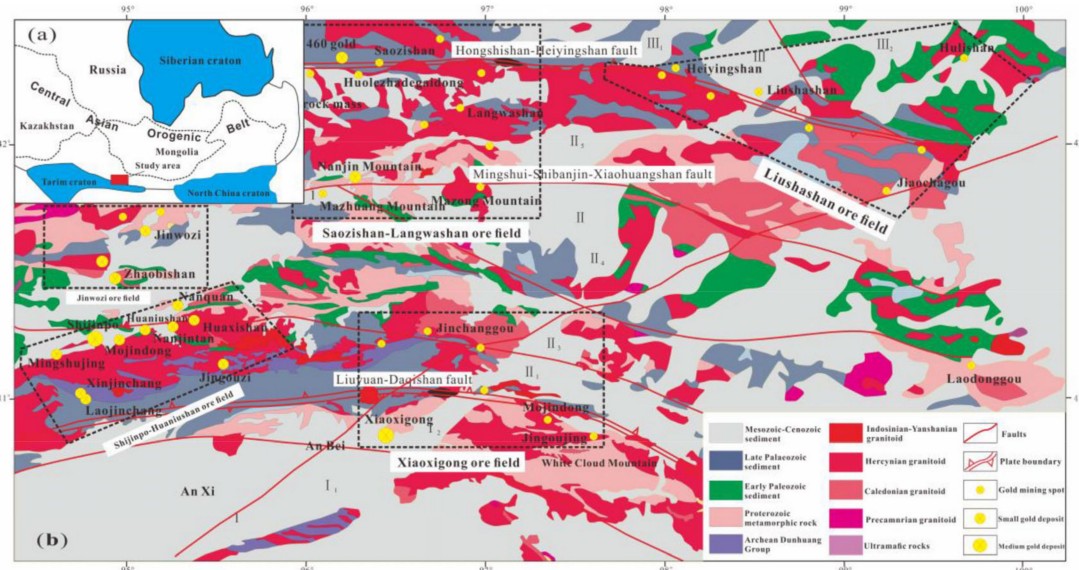

**Figure 1.** (**a**) Geotectonic geological background map of the study area; (**b**) Geological map of Beishan region showing the locations of significant Au deposits.

There is widespread volcanic and magmatic activity in this area, accompanied by severe tectonic movements. Most Au deposits are closely related to Hercynian or Indosinian intrusive rocks [29]. The magmatic activity was mainly concentrated in the Hercynian–Yanshan Period. Movement during the Variscan period was the most prominent in magmatic activity. Along with the migration of magmatic hydrothermal fluid to the structurally weak area, hydrothermal fluid came into contact with the formation to extract significant amounts of elements that are beneficial for mineralisation. Metal elements precipitated at the structural collapse site and formed a series of precious metal mineral resources such as Au, Ag, Pb, and Zn [32].

Overall, the Beishan region has experienced complicated tectonic movements and magmatic activity during its geological evolution and has excellent ore-forming geological conditions. Dozens of Au deposits have been discovered in this area, with the majority clustered in five ore fields: Jinwozi, Shijinpo–Huaniushan, Saozishan–Langwashan, Xiaoxigong, and Liushashan deposits (ranging from west to east and top to bottom) (Figure 1b). The majority of the mineralisation originates from the Carboniferous–Permian (C-P) periods. The outbreak of Au mineralisation occurred during the late Hercynian–Indosinian–Yanshan period in this area [34]. According to observations, the Au occurrences are spatially associated with intrusions. For example, economic ores are generally located in areas that have extensive C-P intrusions [33].

*2.2. Data Preparation*

2.2.1. Spatial Datasets

Spatial data of evidential maps included interpreted rock type faults and Au deposit outcrops collected from the previous research [31]. Geochemical data were acquired from the China Geological Survey Development and Research Center. There were a total of 12,445 stream samples with a sampling grid density of 1 sample per 4 km$^2$. A total of 11 elements were used for modelling (i.e., Ag, As, Au, Bi, Cu, Cr, Hg, Pb, Sb, Sn, and Sr) and were extracted by aqua regia digestion and measured by inductively coupled plasma–mass spectrometry (ICP-MS).

2.2.2. Targets

The binary variable, corresponding to mineral deposit outcrops, was indicated by deposit and non-deposit locations and was labelled as 1 and 0, respectively. A total of 69 known Au occurrences and 69 non-occurrences were taken as training sets, and the latter were selected given the following criteria [14]:

(1)   Selection of non-deposit locations was randomly spatially distributed.
(2)   Non-deposits were distal from any known Au deposits to avoid similar multivariate spatial data characteristics to known mineralisation areas.
(3)   An equal number of non-deposits and deposits were used to balance the number of positive and negative examples, and achieve the optimal model [35].

2.2.3. Predictor Maps

(1)   Intrusive Rock Contact Zone

The contact zone of a rock mass may have a barrier effect, which may indicate the spatial location of Au mineralisation in the Beishan region. Some of the discovered Au occurrences are in the contact zones of intrusive rocks [29]. Therefore, information regarding the proximity to intrusions in terms of the Euclidean distance from the intrusive rocks was extracted (Figure 2a).

(2)   Fault System

Fault density is an ore-controlling variable that may reflect the quantitative characterisation of areas with differential stress changes, or areas of prolonged tectonic activity. High-density fault zones are favoured channels for ore-forming fluid migration and provide possible trapping spaces for mineral precipitation from mineralised fluids. We extracted the faults close to the east–west trending faults/fractures and generated a Euclidean distance map (Figure 2b). Figure 2c shows the linear density map and fault intersection density map. Additionally, a high density of faults is noted on the map of faults based on kernel density (Figure 2d).

(3)   Geochemical Data

A. Local singularity analysis (LSA)
LSA, first proposed by Cheng [36], has been demonstrated to be a useful technique for revealing deep, weak mineralisation in geochemical data that are obscured in the geochemical background. Au, As, Hg, Sb, and Cu were selected in the present study as indicators for the singularity analyses as these metal elements are generally regarded as being related to Au mineralisation. GeoDAS is a graphical Microsoft Windows-based application used to calculate the local singularity indices [37], which is useful to indicate the dispersion and concentration of related geochemical elements and is inversely proportional to the magnitude of the singularity indices. Geochemical element singularity indices maps are shown in Figure 3a–e.

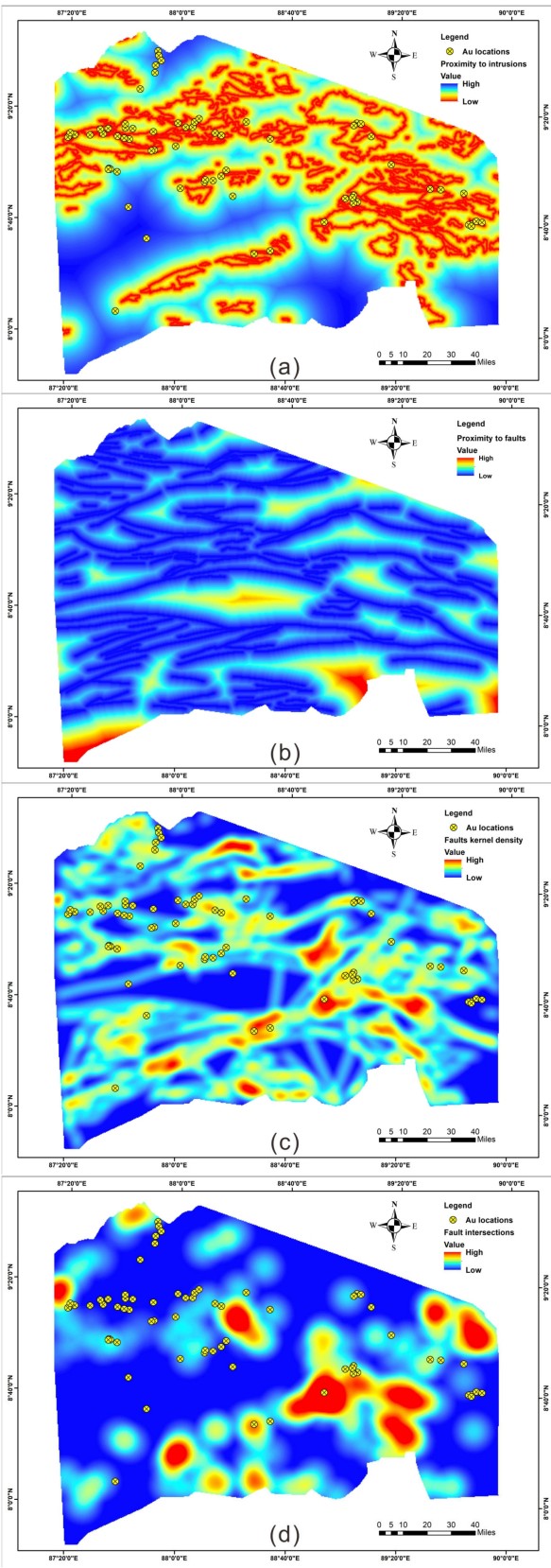

**Figure 2.** Predictor maps of geologic influencing factors. (**a**) Distance to intrusions; (**b**) proximity to faults; (**c**) fault kernel density; and (**d**) faults intersections.

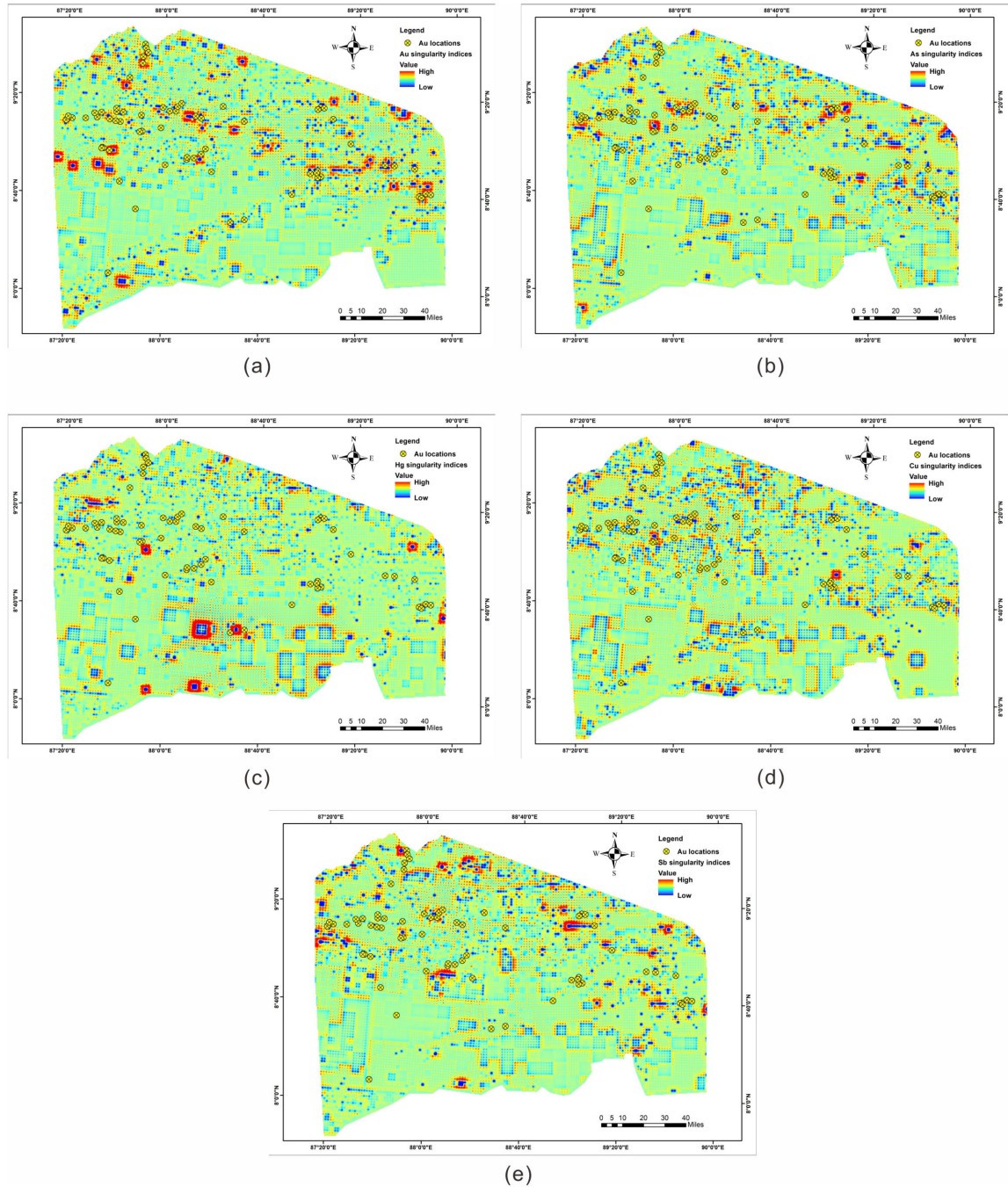

**Figure 3.** Maps of geochemical element singularity indices. (**a**) Au; (**b**) As; (**c**) Hg, (**d**) Cu; and (**e**) Sb.

B. Principal component analysis (PCA)

PCA was employed to extract multi-element geochemical associations that may reflect the signature of the ore-forming processes (Figure 4). From a big data perspective, compared with variables extracted based on mineral systems, principal components (PCs) of these geochemical elements may reveal a correlation with the mineralisation process to a certain extent. Additionally, due to the influence of data closure (i.e., all components sum to a constant), a centred log-ratio transformation method was used to open the closed geochemical data [38].

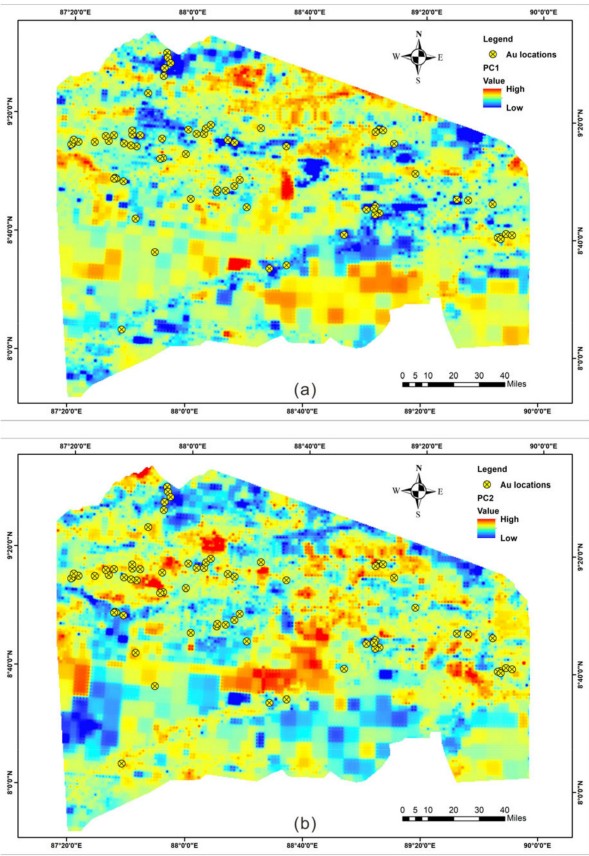

**Figure 4.** Geochemical images of robust principal component analysis. (**a**) PC1; (**b**) PC2.

## 3. Methodology

Figure 5 shows the four necessary primary steps for generating the final prediction maps of mineral prospectivity that were implemented in this study: (1) Preparing a dataset for ensemble spatial modelling; (2) using a relevant coefficient such as Spearman for correlation analysis of features; (3) using RF, SVM, and MaxEnt models and their ensembles for spatial modelling in the MPM; and (4) evaluating models using receiver operator characteristic (ROC) curves and five statistical metrics.

### 3.1. Random Forest

RF is an aggregated predictor with some hierarchical constraints that are used from a root node to a terminal node of each tree to predict the feature represented by the datasets. RF uses a bagging method to randomly select all training subsets, in which every subset forms a decision tree (DT). The diversity of DTs increases by the bagging method to avoid correlations of different DTs [39]. The bagging method randomly resamples the original dataset with a replacement to produce multiple training subsets. As a result, while not utilised in the construction of RF models, approximately one-third of the total instances are employed to validate the prediction accuracy [3]. Therefore, RF can provide a relatively unbiased estimation of the generalisation error without using any other data subset. During the splitting process, DTs search through the optimal split, which can be measured by the maximum reduction in impurity. There are many approximations for impurity measurements, with the most common measure being the Gini index, shown by the following equation (Equation (1)):

$$I_G t_{Y(x_i)} = 1 - \sum_{j=1}^{m} f\left(t_{Y(x_i)}, \; j\right)^2 \tag{1}$$

where $f\left(t_{Y(x_i)},\ j\right)$ is the probability of samples for which the value $x_i$ belongs to the leaf $j$ and is the node $t(x)$; $Y$ is the search value of the splitting process; $m$ denotes the number of trees. The DT which splits the criteria selection is based on the lowest Gini impurity index (IG).

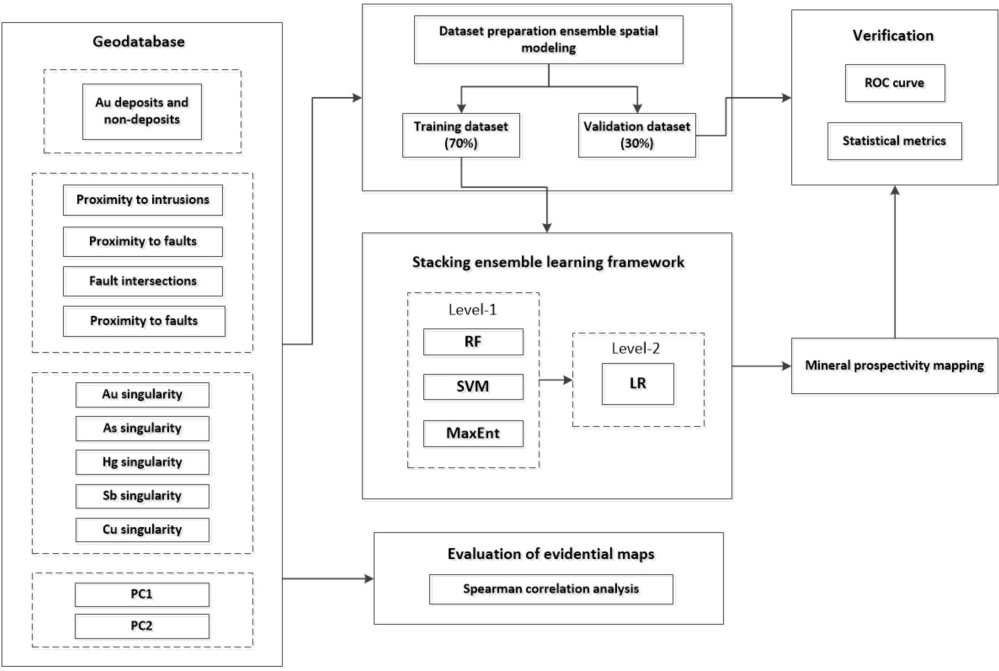

**Figure 5.** Flowchart of this study. RF—random forest; SVM—support vector machine; MaxEnt—maximum entropy model; LR—logistic regression.

The final result of RF modelling is determined by the average of all DT predictions, as shown in Equation (2) [40]:

$$f_{rf}^{K}(x) = \frac{1}{K}\sum_{k=1}^{K} T(x) \tag{2}$$

where $f_{rf}^{K}(x)$ represents the result of the RF regression predictor; $x$ denotes the input vector; $K$ denotes the number of regression trees ranging from 1 to $k$; $T(x)$ denotes the result of the DT prediction.

### 3.2. Support Vector Machine

The SVM algorithm, commonly used for solving problems in regressions, comprises a heuristic algorithm based on statistical methods [41]. In complicated non-linear cases, the SVM method converts input training instances into a higher-dimensional space, where regression calculations can be linearly separated. After the most optimal classifier is determined, the new dataset with unknown category information can be predicted by the trained SVM model, based on the characteristics of the training dataset [42].

The choice of a kernel function and its parameters for an SVM is crucial for achieving reliable results. The objective function is as follows:

$$\mathrm{F}(x) = \mathrm{sgn}(\omega x + b) \tag{3}$$

where sgn is a sign function, which is defined as:

$$\mathrm{sgn}(x) = \begin{cases} 1 \text{ if } x > 0 \\ 0 \text{ if } x = 0 \\ -1 \text{ if } x < 0 \end{cases} \tag{4}$$

Moreover, $\omega$ and $b$ are the two parameters that were used to separate the hyperplane decision function, which was obtained by determining the optimisation function as follows:

$$\tau(w) = \frac{1}{2} \|w\|^2 \tag{5}$$

subject to:

$$y_i((wx_i) + b) \geq 1, \ i = 1, \ldots \ldots, l \tag{6}$$

The key to solve this optimisation problem is the saddle point of the Lagrange function:

$$L(w, b, \alpha) = \frac{1}{2}\|w\|^2 - \sum_{i=1}^{l} \alpha_i(y_i((x_i w) + b) - 1) \tag{7}$$

$$\frac{\partial}{\partial b}L(w, b, \alpha) = 0, \ \frac{\partial}{\partial w}L(w, b, \alpha) = 0 \tag{8}$$

where $\alpha_i$ is a Lagrange multiplier. The Lagrange function is minimised for $\omega$ and $b$ and is maximised concerning $\alpha_i > 0$.

The sigmoid kernel function is as follows:

$$K(x_i, x_j) = \tanh(\lambda x_i x_j + r), \ \lambda > 0 \tag{9}$$

where the parameter $\lambda$ serves as an inner product coefficient in the hyperbolic tangent function, and $\gamma$ is used for kernels of polynomial and sigmoid types. Further details of the reasoning behind the mathematical formula of SVM algorithms have been provided by the previous research [14].

### 3.3. Maximum Entropy Model

The MaxEnt model originated from the statistical learning theory [43], which was later developed to solve the problem of the geographical distribution of species [44]. It can be interpreted from the perspective of machine learning [45], and its applicability to MPM has previously been demonstrated [46]. The principle of maximum entropy could be used to quantify the spatial relationship between known deposits and ore-controlling variables, which perform numerous iterations based on the most critical features until the overall accuracy converges and reaches the optimum [47]. Some function types (such as linear, quadratic, and threshold) can be used to fit highly complex response functions to characterise the distribution trend of current data through integration quantitatively. Detailed reasoning behind the mathematical formula is provided in a previous study [46].

The MaxEnt model has the following merits:

(1)  It is possible to integrate both discrete and continuous variables, and the output result provides a minimum deviation estimate of the target distribution [43];
(2)  It requires only presence data, along with evidential maps for the study area [44];
(3)  It can apply regularisation parameters to reduce the risk of data overfitting (which determines the error bounds around the mean of the observed data) and optimise the function to fit the data distribution trends [46].

### 3.4. Ensemble Learning Method Framework

Recently, modelling has received widespread attention because of its ability to solve high-dimension problems and improve the predictive performance of individual algorithms [48]. Bagging and boosting were the most representative homogeneous ensemble methods when they were initially proposed [49], whereas stacking became a standard technique for the heterogeneous ensemble method.

Unlike the other two types of ensemble learning methods, SELM uses a meta-learning technique to combine different types of algorithms. The structure of the SELM framework consists of two levels.

The first level is similar to a highly complex non-linear variable converter, and then, the instances have new features after this conversion; therefore, the second level does not require complex classifiers. Logistic regression (LR) was recommended by previous researchers [24]. The process of the stacking method is as follows:

(1) The base algorithms are trained by using k-fold cross-validation (usually k = 5 or 10) on the same datasets.
(2) The three base models with remarkable performances are selected to provide predictions, and the k-fold cross-validation is also used.
(3) The mean value of the three base learners' k-fold cross-validation results are regarded as the new representations.
(4) A senior model can be trained given the new representations.

*3.5. Model Evaluation Metrics*

The MPM's performance was comprehensively assessed by an ROC curve and five statistical metrics. A confusion matrix can accurately explain the resulting predictive models. Based on the confusion matrix, the five statistical metrics were computed to evaluate the performances of the different models. The formulas are as follows [23,50]:

$$\text{Accuracy} = \frac{TP + TN}{TP + TN + FP + FN} \tag{10}$$

$$\text{Recall} = \frac{TP}{TP + FN} \tag{11}$$

$$\text{Precision} = \frac{TP}{TP + FP} \tag{12}$$

$$\text{F} - \text{measure} = \frac{2TP}{2TP + FP + FN} \tag{13}$$

$$\text{Kappa index} = \frac{TP + TN - [(TP + FN)(TP + FP) + (FP + TN)(FN + TN)]/(TP + FP + FN + TN)}{(TP + FP + FN + TN) - [(TP + FN)(TP + FP) + (FP + TN)(FN + TN)]/(TP + FP + FN + TN)} \tag{14}$$

In Equations (10)–(14), the true-positive (TP) sample is a deposited sample that is correctly classified as a "deposit"; a false-negative (FN) sample is a deposited sample that is incorrectly taken as a "non-deposit"; a true-negative (TN) sample is a non-deposit sample that is incorrectly classified as a "deposit". The ROC curve is generated by illustrating the true-positive rate (TPR) on the *y*-axis to the false-positive rate (FPR) on the *x*-axis. The closer the ROC curve becomes to the upper left corner, the better the model performs [51].

## 4. Experiments and Results

*4.1. Experiments*

Model stacking is divided into two steps. In the first step, the predictions of three base learners are taken as new features for building a senior model. Furthermore, the robustness of the new representations as training sets is ensured by using five-fold cross-validation. In the second step, LR is usually selected as the meta-model to construct the model and obtain a final prediction. The training process of the SELM is shown in Figure 6.

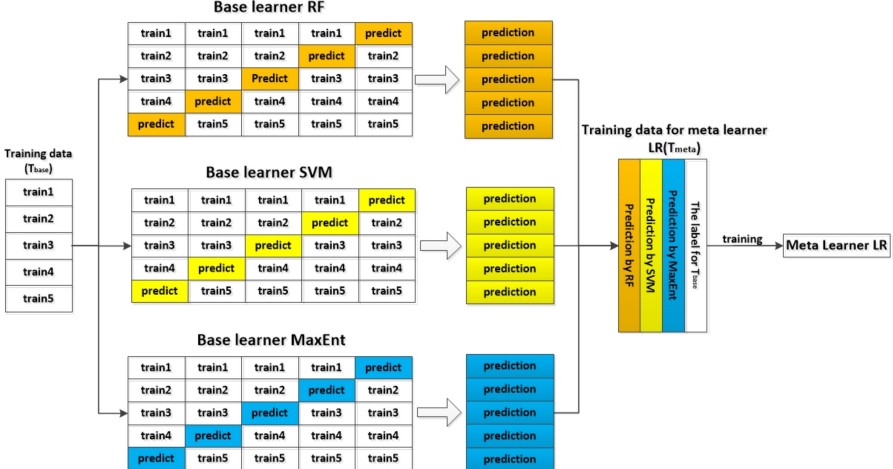

**Figure 6.** The training process of stacking ensemble learning method framework.

### 4.1.1. Data Processing

Preparation of the input datasets (including features and targets) is the first step of MPM. The integration of multi-source datasets was implemented by using ESRI's ArcGIS 10.5 software. The SELM framework, including model training and evaluation, was implemented under the framework of the scikit-learn package and in Python 3.6 of Pycharm, which provides rich resources of machine learning algorithm packages. Furthermore, every evidence raster layer was created with a cell size of 500 m. This grid cell size was determined depending on the spatial distribution pattern of known gold deposits and related faults and intrusions that ensure only one deposit exists in any given grid unit [52].

The known occurrences were randomly divided for spatial modelling in this study area, given a general sampling strategy [1,27], i.e., 70% of occurrences (97) for training and the remaining 30% of occurrences (41) for testing. Predictors are regarded as critical conditions for mineral potential prediction. We extract eleven evidential maps which are related to geological and geochemical information as datasets, based on the controlling features of Au mineralisation described in Section 2.

### 4.1.2. Training Base Models

The given algorithms' parameter combinations of the ensemble learning method framework were traversed by using grid search, and the optimal parameter combination was determined through 10-fold cross-validation in the scikit-learn package. Model training is an important step to produce accurate predictions. The first step is to identify the optimal parameters. Empirical knowledge may be useful in this research; however, a trial-and-error procedure is required to gain the optimal configuration of the parameters. Ten-fold cross-validation was used to evaluate the predictions. The training dataset was randomly divided into 10 equal subsets, 9 of which were used to train the model, and the remaining set was used as the validation dataset. This procedure was undertaken 10 times with various subsets that served as the validation dataset in turns [53].

After training the prediction models, spatial models were constructed in the form of probability grids in the ArcGIS environment. Each grid cell in the map was assigned a prospecting index, which represents the probability of the existence of deposit output by the corresponding method.

### 4.1.3. Constructing Mineral Prospectivity Maps

For better visualisation and understanding of the overall spatial pattern of the distribution of Au deposits, the probability was reclassified into four levels: low, moderate, high, and very high by using the concentration–area (C-A) fractal method. Then, mineral prospectivity maps by different methods

were obtained, as shown in Figure 7; Figure 8 shows the percentages of different levels in different methods and their ensembles.

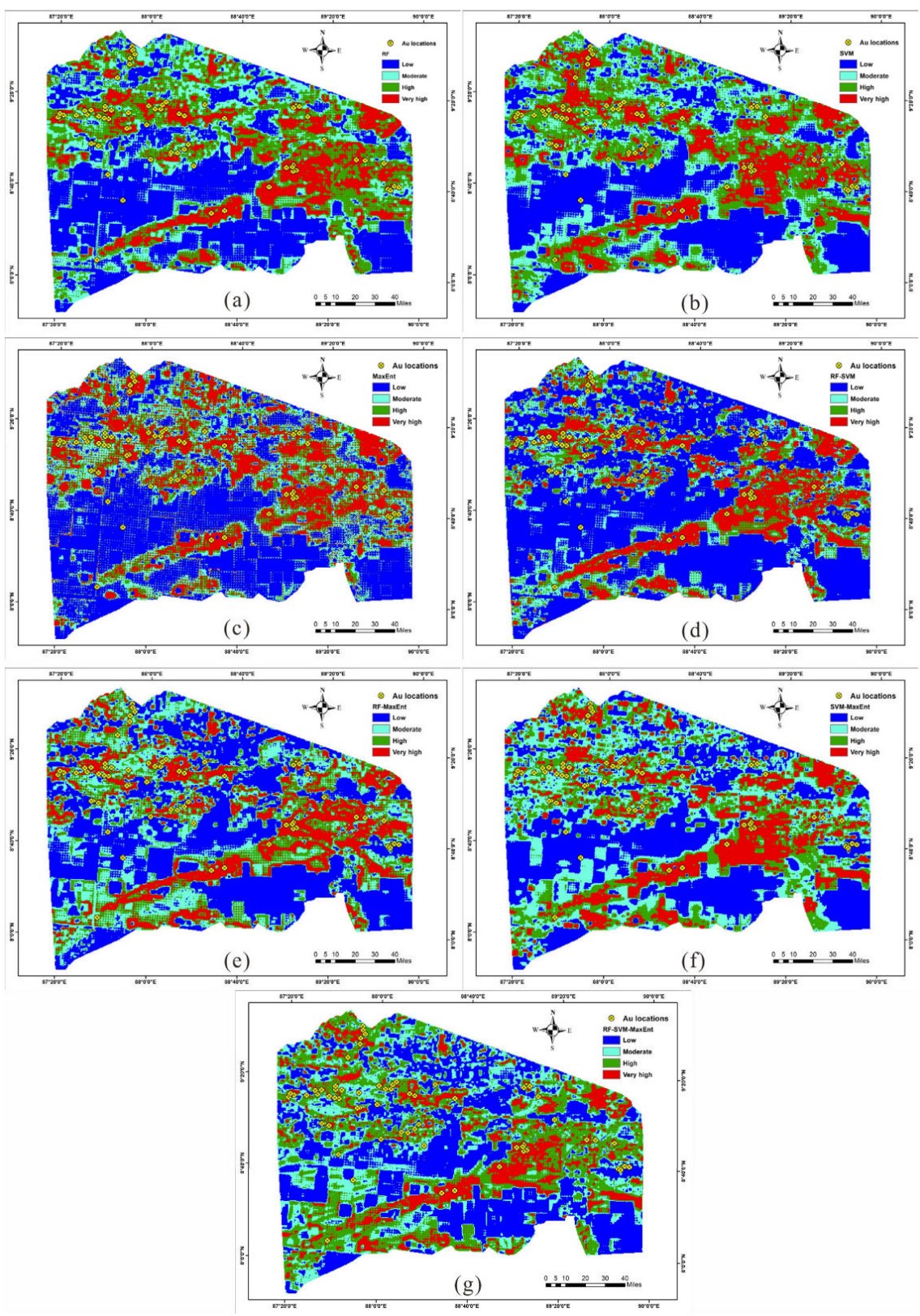

**Figure 7.** Mineral prospectivity maps by different methods: (**a**) RF; (**b**) SVM; (**c**) MaxEnt; (**d**) RF–SVM; (**e**) RF–MaxEnt; (**f**) SVM–MaxEnt; (**g**) RF–SVM–MaxEnt.

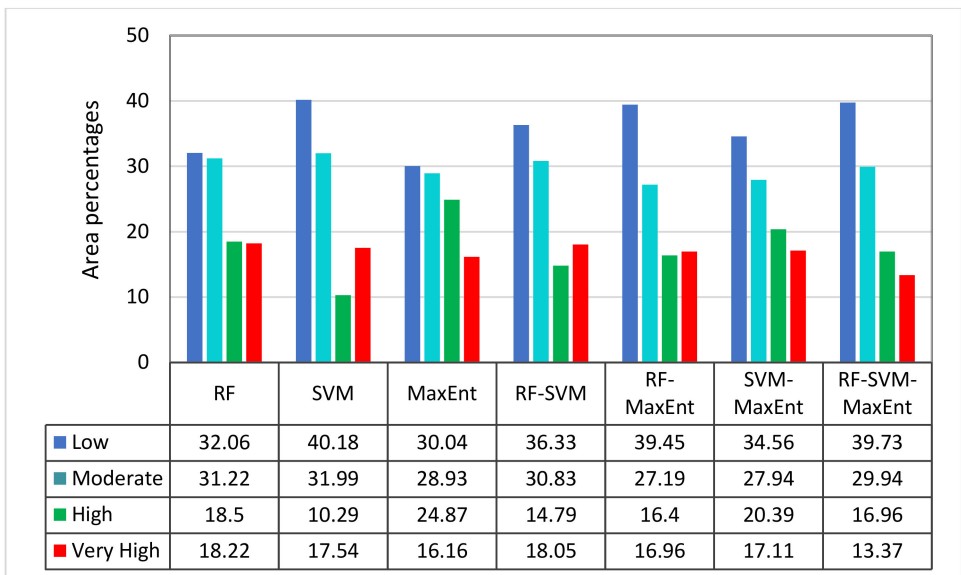

**Figure 8.** Percentages of different mineral prospectivity prediction classes.

According to observations, the spatial distribution maps of mineral prospectivity generated by different techniques and their ensembles in the framework share some similar rules to some extent. For example, relatively high mineralisation potential areas are mainly distributed in the north and east of the study area, while other regions have relatively low probability potential. Moreover, the historically known deposits are mostly located in very high and high potential areas. Among them, RF–SVM–MaxEnt occupied 13.17% of the very high potential in this study area, while high, moderate, and low potential areas accounted for 16.96%, 29.94%, and 39.73% of the study area, respectively. Furthermore, compared with a single model, the spatial distribution of potential Au deposits based on the SELM framework is more aligned with known deposits.

### 4.2. Comparison of Prediction Results

Mineral prospectivity modelling has no scientific significance if the validity of the results is not calculated [54]. Assessing the data's fit and the predictive ability of the methods for the training data is the first stage of validation. Subsequently, validation is performed through the testing data, i.e., to evaluate the prediction rate of the model built with the training data. Statistical metrics were developed for quantifying the overall performance of individual models and their ensembles [55]. Performances of the different methods using the test dataset are presented in Table 1.

The ensemble learning methods of RF–SVM–MaxEnt showed better performance than the base classifiers, and it achieved the highest overall accuracy value of 0.928, followed by RF (0.910), RF and SVM (0.901), SVM and MaxEnt (0.859), SVM (0.856), RF and MaxEnt (0.829), and MaxEnt (0.823), respectively.

**Table 1.** Prediction accuracy of different methods on the validation.

| Methods | Performance | | | | |
|---|---|---|---|---|---|
| | Accuracy | Recall | Precision | F-measure | Kappa |
| RF | 0.910 | 0.870 | 0.940 | 0.904 | 0.819 |
| SVM | 0.856 | 0.944 | 0.800 | 0.864 | 0.713 |
| MaxEnt | 0.823 | 0.926 | 0.702 | 0.730 | 0.798 |
| RF–SVM | 0.901 | 0.926 | 0.877 | 0.901 | 0.802 |
| RF–MaxEnt | 0.829 | 0.870 | 0.797 | 0.832 | 0.708 |
| SVM–MaxEnt | 0.859 | 0.907 | 0.817 | 0.860 | 0.723 |
| RF–SVM–MaxEnt | **0.928** | **0.907** | **0.942** | **0.925** | **0.856** |

## 5. Discussion

### 5.1. Prediction Performance of Different Methods

Mineralisation is a relatively complex process. To date, a few scholars have explored new techniques for accurate prediction [19,22,56]. Because of the high complexity of metallogenic prediction and the uncertainty of multiple variables in the modelling process, the prediction methods are limited to some extent. Moreover, the predictive performance of potential zones could be evaluated by an ROC curve. Specifically, the ROC curve is obtained by drawing all the sensitivity combinations and false-negative (1-specificity) ratios on the *y*-axis and *x*-axis, respectively, which may be gained by varying the decision threshold. Sensitivity is the proportion of positive cases of deposits that are correctly predicted, while 1-specificity refers to the proportion of occurrences that are not mispredicted [57]. Given the area under the curve (AUC), the ROC drawing could be quantitatively summarised, which ensures the accuracy of the model for the prediction of mineral prospectivity. Furthermore, the AUC value ranges from 0 to 1; the higher the value, the higher the prediction rate, while a value close to 0.5 indicates that the predictive ability is not better than a random guess [58].

Based on the experimental results, RF–SVM–MaxEnt achieved the best performance, followed by SVM, SVM–MaxEnt, RF–MaxEnt, RF–SVM, RF, and MaxEnt. The graph of the ROC curves and AUC are shown in Figure 9. And it can be observed that RF–SVM–MaxEnt had the highest AUC value of 0.985, followed by RF–SVM (0.977), RF (0.965), SVM (0.965), SVM–MaxEnt (0.963), and MaxEnt (0.900). These results show that MPM has a much larger AUC value in the ensembles, indicating that heterogeneous SELM will help to reduce the uncertainty of multiple variables to some extent in mineral potential modelling.

In this article, the SELM framework is applied to integrate three heterogeneous machine learning methods (RF, SVM, MaxEnt) through a meta-learner logistic regression (LR). Subsequently, this study compared the base models and ensembles (RF–SVM, RF–MaxEnt, RF–SVM–MaxEnt) in MPM. Specifically, the predictive performance of RF and SVM is better than MaxEnt in this study area. Moreover, this finding is in accordance with previous research [27,50]. Furthermore, RF–SVM–MaxEnt is superior to any base model and other ensembles according to the experimental results. This may be explained by the fact that if the base model mistakenly learns a particular region in the feature space and this results in a misclassification, the meta-learner may classify it correctly at the second layer when provided with other learners in the SELM framework [24]. Furthermore, conventional heterogeneous algorithms give more potential and possibility for spatial modelling, which may be effectively used for the prediction of MPM. In this article, SELM can be taken as a promising technique for MPM, due to it showing improved predictive performance against their components. Generally, several base models by the ensemble can obtain better performance [59]. However, the availability requires to be further investigated for mineral potential modelling. Furthermore, SELM achieved better predictive performance than other base models in a recent study [24,26]. In this previous relevant research, the merits of SELM have been further proved.

### 5.2. Importance of Variables

The feature importance measures (including influencing geologic features, geochemical singularity indices, and PCs) can be quantified based on the contribution of each feature to the final prediction by the different base models (Figure 10); therefore, heterogeneous modelling results may be interpreted to some extent.

In the RF modelling process, the relative importance of the variables showed that advanced spatial analytical techniques are potent methods for capturing and extracting ore-forming information, such as Hg, As, and Au singularity indices [36], followed by proximity to intrusions, fault kernel density, and Cu singularity indices. Other features did not contribute significantly as influencing factors in the MPM modelling. It is worth pointing out that Hg singularity indices impose a significant influence on predictive models, and that such influence was even more significant than the influences

exerted by proximity to intrusions and regional structures, which are well-recognised ore-controlling factors. In the SVM modelling process, proximity to intrusions made the most significant contribution to the final predictive model, followed by PC1, faults kernel density, and Au, Hg, and Cu singularity indices. Other features did not contribute significantly as influencing factors in the MPM modelling. Ore-forming materials of intrusions played a vital role in the formation of Au ores and should be regarded as an essential exploration criterion. The evidential layer with the highest contribution to the MaxEnt model was the Au singularity indices, and the other features made almost the same significant contribution to the final prediction to MPM. Feature importance measures indicated that heterogeneous SELM might take full advantage of all ore-controlling features from multiple dimensions to reduce the uncertainty of multiple variables as soon as possible in mineral potential modelling. Above all, the importance of factors can provide insights into the different models of Au mineralisation to some extent in this study area.

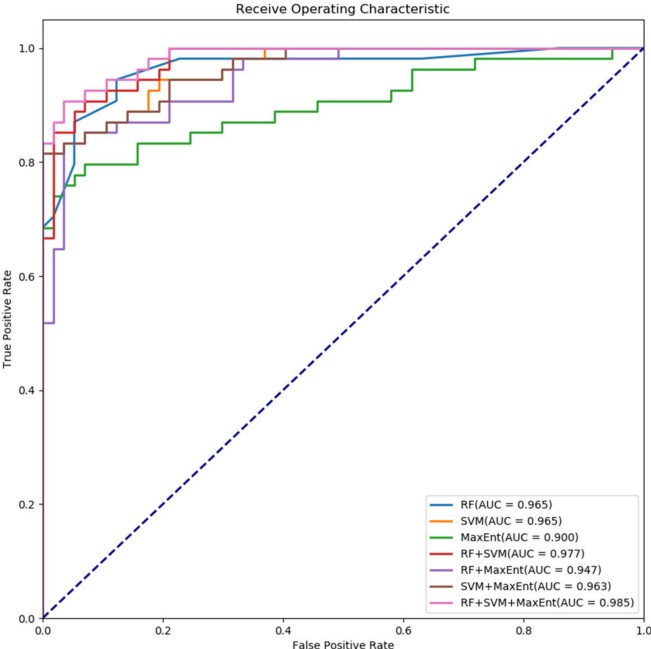

**Figure 9.** ROC curves and AUC analysis by different predictive models.

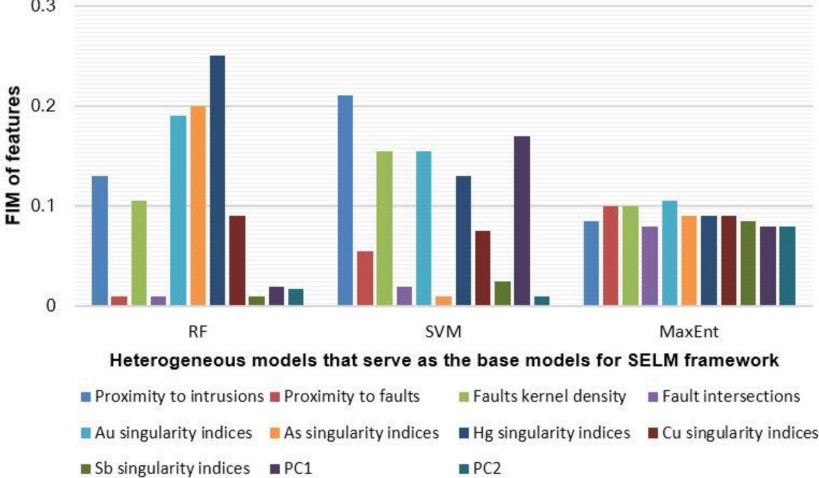

**Figure 10.** Feature importance measures (FIMs) of mineralisation influencing factors using different models.

*5.3. Reliability from a Geological Perspective*

From a geological perspective, the primary outcrops comprise the Precambrian crystalline bedrock and overlying sedimentary lithologies of the Palaeozoic–Mesozoic. These intrusions were formed from the Precambrian–Permian and were separated by well-developed fault-related hosts and grabens [32]. Meanwhile, these tectonic units were affected by multi-stage magma intrusion during the Carboniferous and Permian. The fault structure is well developed, including EW-, NNE-, and NNW-NE-trending basement faults. The structure in the study area is relatively complicated, with large-scale strong tectonic, magmatic, and metamorphic effects, which provides heat and favourable channels for the activation, transfer, and enrichment of metal elements, and finally, forms abundant Au resources [33]. In this research, the MPM gained by the SELM framework has adequate adaptability and practicability, and the zones with very high and high metallogenic probability account for less than one-third of the study area, which is primarily distributed in the zones which are close to the intrusions from Precambrian to Permian. Furthermore, most deposits in the predictive zones are located along two deep-seated faults of Hongshishan–Heiyingshan and Liuyuan–Daqishan.

Based on the previous study [29], there is a strong spatial correlation between gold deposits and intrusive bodies. Furthermore, most of these deposits are located along two deep-seated faults of Hongshishan–Heiyingshan and Liuyuan–Daqishan [60–62]. Therefore, the previous research proved our final prediction result. It may be explained that the ore-forming materials from the Precambrian metamorphic rocks and late Palaeozoic volcanic rocks may be transferred along with the favourable structures when they went up [34,61,62]. Additionally, some gold deposits may be formed because of the remobilisation effect of several deep-seated faults, which causes the emplacement of granitoid. Therefore, it is believed that there are still substantial gold potentials in the Beishan region, due to Hercynian magmatism related to the formation of gold deposits being quite intense.

## 6. Conclusions

We propose a SELM framework to explore the potential of ensemble learning to improve predictive performance for MPM. According to the experimental results, we can draw the following conclusions:

Prospective areas showed by MPM were aligned with known Au deposits in the spatial pattern, and the RF–SVM–MaxEnt ensemble learning method is remarkably higher than the base classifiers in the potential application of MPM. Therefore, the SELM framework is a promising technique to improve generalisation accuracy in mineral potential modelling.

There are some limitations in this research, and further work is required to be implemented in the future. Firstly, more reliable data, such as remote sensing data and geophysical data, should be considered and applied for mineral potential modelling in the next work. Secondly, each base model can be enhanced. Deep learning techniques may outperform the other machine learning methods in mineral potential modelling. Our research will need to focus on exploring more novel effective deep learning and ensemble learning techniques in the application of MPM.

Lastly, more detailed geological surveys and explorations are essential in future work. Specifically, engineering geology and 3D modelling techniques should be taken into account. Furthermore, for the very high potential zones, engineering measures are recommended, such as drilling, geophysical techniques, and constructing underground 3D geological models for further accurate targeting.

**Author Contributions:** Conceptualisation, X.Z.; methodology, K.W.; investigation, K.W.; validation, X.Z., G.W. and K.W; resources, X.Z. and G.W.; data curation, N.C.; writing—original draft preparation, K.W.; visualisation, X.Z. and K.W.; funding acquisition, X.Z., and D.L. All authors have read and agreed to the published version of the manuscript.

**Funding:** This research was funded by the National Natural Science Foundation of China (Grant No. 72033005), the Fundamental Research Funds for the Central Universities (Grant No. 2652019001 and No. 2652020004), and China Geological Survey Development and Research Center (Grant No. D21705601 and No. D218071). Acknowledgements: We are grateful to Shuai Zhang, Fan Yang, Tao Sun, the handling editors and two anonymous reviewers for their constructive comments and suggestions that significantly improved the quality of the article.

**Conflicts of Interest:** The authors declare no conflict of interest.

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
