# Peer review of "A Multi-Model Ensemble Approach for Gold Mineral Prospectivity Mapping: A Case Study on the Beishan Region, Western China"

_minerals, doi:10.3390/min10121126_

Round 1

Reviewer 1 Report

Dear Authors,

Thanks for submitting your research to Minerals.

This research sounds interesting to me. I suggest you have one more quick look at it for spell checking. Also in some cases, you have not removed the references. I mean still the name of the authors are next to the numbers in the manuscript!

Also, after several times talking about MPM, finally, you have clarified what MPM is. You need to do it at first.

The paper is generally good, but it is not plain and straightforward enough for general geoscientists. I mean your readers should be experts in this field to be able to understand your paper well and apply the methods. Therefore, my biggest concern in your readers. Imagine a classic geologist needs to understand this paper too to be encouraged to try the methods. If I was in your shoes, I would simplify the presentation. Of course, I know it is not always easy, but the main aim we have in science is to make it FAIR (i.e., Findable, Accessible, Interoperable, and Reusable). It is up to you to simplify the scientific language you have used though.

All in all, this paper looks acceptable to me, after those mentioned minor revisions.

Best,

Author Response

Point 1: “I suggest you have one more quick look at it for spell checking. Also, in some cases, you have not removed the references. I mean still, the name of the authors are next to the numbers in the manuscript!”

Response 1: Thank you for your reminding. We have removed the references, which the name of the authors are next to the numbers in the manuscript according to your suggestion. Thanks very much.

Point 2: “Also, after several times talking about MPM, finally, you have clarified what MPM is. You need to do it at first.”

Response 2: Thank you for your reminding. We have clarified what MPM is in Line 16 at first. Thanks a lot.

Point 3: “The paper is generally good, but it is not plain and straightforward enough for general geoscientists. I mean your readers should be experts in this field to be able to understand your paper well and apply the methods. Therefore, my biggest concern in your readers. Image a classic geologist needs to understand this paper too to be encouraged to try the methods. If I was in your shoes, I would simplify the presentation. Of course, I know it is not always easy, but the main aim we have in science is to make it FAIR (i.e., Findable, Accessible, Interoperable, and Reusable). It is up to you to simplify the scientific language you have used though.”

Response 3: Thank you for your valuable suggestion. We have revised the sections of abstract, introduction, experiments and results, discussion, and conclusion in language presentation, and hopefully, it will be easier for the reader to understand. Besides, we have invited the polishing company to help to modify the problems of language in order to avoid redundancies and keep it short as soon as possible.

Once again, thank you very much for your constructive comments and suggestions which would help us both in English and in-depth to improve the quality of the article.

Reviewer 2 Report

In the present manuscript, authors have tried to analyse the potential application of the stacking ensemble learning methods for mineral prospectively mapping with emphasis to gold ore in Beishan region, western China. The paper needs further revision on the structure, data analysis front along with discussion. The article is poorly written without much attention to the discussion and request to focus on this. The article needs substantial revision prior to publication, and few of the comments has listed for the reference.

  • Abstract: Be specific and crisp without mentioning many values. It must attract the reader to read the full manuscript. Please revise. The abstract needs to be shortened with the central theme of the research by highlighting the novelty aspects.
  • The paper depends on the results of the few simulation tests without much focus on DOE, or repeatability.
  • Better connect your research findings to previous works published in the exploration research area.
  • The main drawback of this paper is the lack of proper discussion on simulation results. The discussion section needs to be strengthened with the available literature and the innovative way of analysis achieved.
  • The innovation and the importance of this work are not highlighted in the abstract, introduction and conclusions.
  • The manuscript needs a thorough revision of its language and style. Overall, this paper is challenging to read. Avoid redundancies and keep it short. I suggest a thorough overhaul of the text for a more precise understanding of the reader.
  • Introduction: This section needs to expand further on the mineral prospectivity mapping with proper citations.
  • Introduction: Author should discuss the problem associated with current mapping practice, and it’s importance along with literature (with citation).
  • Introduction: There are many articles on this subject. The author should be explicit the novelty of the current paper along with the gap area.
  • Introduction: Author should narrate the merits and demerits of prospectivity mapping methods.
  • Materials and Methods: Why authors have selected Beishan region for the present research?
  • Conclusion: These are observations not conclusions/recommendations. This section needs to be crisp and a summary of the findings rather than discussion. Please revise it accordingly. Also, request not use bullet point in this section.
  • References: There are many articles on this subject. The author needs to review all of these published articles.

Author Response

Point 1: The paper needs further revision on the structure, data analysis front along with discussion.

Response 1: Thank you for this suggestion, we have revised and merged the Section.4 and section. 5 according to your comments, where we added the data analysis in Line 317-322, and Line 365-367; Also we added the contents of discussion in Line 386-410, Line 429-432, and Line 446-459.

Point 2: The article is poorly written without much attention to the discussion and request to focus on this.  

Response 2: We appreciate the comments. We have focused on revising the discussion section based on your suggestion. Therefore, we have changed two subsections into three subsections and added more relevant content in every subsection. We added the discussion in Line 374-375, Line 384-408, Line 427-430, and Line 444-457.

Point 3: Abstract: Be specific and crisp without mentioning many values. It must attract the reader the full manuscript. Please revise. The abstract needs to be shortened with the central theme of the research by highlighting the novelty aspects.

Response 3: Thank you for the comments. We have highlighted the central theme at the beginning of the article from Line 16-21 for attracting the reader the full manuscript according to your suggestion.

Point 4: The paper depends on the results of the few simulation tests without much focus on DOE, or repeatability.

Response 4: We appreciate the comments. We have added the experiments in section. 4 based on your suggestion. The revision content in Line 300-350.

Point 5: Better connect your research findings to previous works published in the exploration research area.

Response 5: Thank you for your suggestion. We have added some previous works published in the discussion section. The literatures of [29], [34], [60], [61], and [62] were cited in subsection. 5.3, which proved our research findings. The revision content in Line 444-457.

ref:

[29] Jiang, S.; Nie, F.; Liu, Y., Gold Deposits in Beishan Mountain, Northwestern China. Resource Geology 2004, 54, (3), 325-340.

[34] Yue, Y.; Liou, J. G.; Graham, S. A., Tectonic correlation of Beishan and Inner Mongolia orogens and its implications for the palinspastic reconstruction of north China. Geological Society of America Memoirs 2001, 194, 101-116.

[60] Zhang, M., Analysis on the causes and prospecting direction of gold deposits in North Mt. area of Gansu Province (in Chinese). World Nonferrous Metals 2019, (7), 297-298.

[61] Jiang, S.; Nie, F., Geology and ore genesis of the Nanjinshan gold deposit in Beishan Mountain area, northwestern China. Mineral Deposit Research: Meeting the Global Challenge 2005, 537-540.

[62] Liu, X.; Wang, Q., Tectonics of orogenic belts in Beishan Mts., western China and their evolution (in Chinese). Geosci. Research 1995, 28, 37-48.

Point 6: The main drawback of this paper is the lack of proper discussion on simulation results. The discussion section needs to be strengthened with the available literature and the innovative way of analysis achieved.

Response 6: We appreciate the comments. We have added the proper discussion on simulation results in Line 384-408 and Line 444-457. We have cited the available literatures, such as [24], [26], [27], [29], [34], [50], [59], [60], [61], and [62] for strengthening this section. Also, we added one subsection, named “Importance of variables”, to discuss the possible reason of simulation results further.

ref:

[24] Zhang, Y.; Li, M.; Han, S.; Ren, Q.; Shi, J., Intelligent Identification for Rock-Mineral Microscopic Images Using Ensemble Machine Learning Algorithms. Sensors (Basel) 2019, 19, (18).

[26] Fang, Z.; Wang, Y.; Peng, L.; Hong, H., A comparative study of heterogeneous ensemble-learning techniques for landslide susceptibility mapping. International Journal of Geographical Information Science 2020, 1-27.

[27] Zhang, S.; Xiao, K.; Carranza, E. J. M.; Yang, F., Maximum Entropy and Random Forest Modeling of Mineral Potential: Analysis of Gold Prospectivity in the Hezuo–Meiwu District, West Qinling Orogen, China. Natural Resources Research 2018, 28, (3), 645-664.

[29] Jiang, S.; Nie, F.; Liu, Y., Gold Deposits in Beishan Mountain, Northwestern China. Resource Geology 2004, 54, (3), 325-340.

[34] Yue, Y.; Liou, J. G.; Graham, S. A., Tectonic correlation of Beishan and Inner Mongolia orogens and its implications for the palinspastic reconstruction of north China. Geological Society of America Memoirs 2001, 194, 101-116.

[60] Zhang, M., Analysis on the causes and prospecting direction of gold deposits in North Mt. area of Gansu Province (in Chinese). World Nonferrous Metals 2019, (7), 297-298.

[61] Jiang, S.; Nie, F., Geology and ore genesis of the Nanjinshan gold deposit in Beishan Mountain area, northwestern China. Mineral Deposit Research: Meeting the Global Challenge 2005, 537-540.

[62] Liu, X.; Wang, Q., Tectonics of orogenic belts in Beishan Mts., western China and their evolution (in Chinese). Geosci. Research 1995, 28, 37-48.

Point 7: The innovation and the importance of this work are not highlighted in the abstract, introduction and conclusions.

Response 7: Thank you for your helpful advice. We have highlighted our innovation and the importance of this work in abstract, introduction and conclusion according to your suggestion. The revision content in Line 16-21, Line 7-85,and Line 463-465.

Point 8: The manuscript needs a thorough revision of its language and style. Overall, this paper is challenging to read. Avoid redundancies and keep it short. I suggest a thorough overhaul of the text for a more precise understanding of the reader.

Response 8: Thank you for your suggestion. We have thoroughly revised language and style in our manuscript. Moreover, we have invited the polishing company to help to modify the problems of language in order to avoid redundancies and keep it short as soon as possible.

Point 9: Introduction: This Section needs to expand further on the mineral prospectivity mapping with proper citations.

Response 9: We appreciate the comments. We have expanded further on the mineral prospectivity mapping with proper citations in the introduction section, such as [11], [12], [18], [19], [23], [24], [26], [27], [20], [28], [29], and [30]. The revision content in Line 45-50, Line 52-55, Line 62-65, Line 70-73, and Line 75-85.

ref:

[12] Porwal, A.; Carranza, E. J. M., Introduction to the Special Issue: GIS-based mineral potential modelling and geological data analyses for mineral exploration. Ore Geology Reviews 2015, 71, 477-483.

[20] Shabankareh, M.; Hezarkhani, A., Application of support vector machines for copper potential mapping in Kerman region, Iran. Journal of African Earth Sciences 2017, 128, 116-126.

[26] Fang, Z.; Wang, Y.; Peng, L.; Hong, H., A comparative study of heterogeneous ensemble-learning techniques for landslide susceptibility mapping. International Journal of Geographical Information Science 2020, 1-27.

[27] Zhang, S.; Xiao, K.; Carranza, E. J. M.; Yang, F., Maximum Entropy and Random Forest Modeling of Mineral Potential: Analysis of Gold Prospectivity in the Hezuo–Meiwu District, West Qinling Orogen, China. Natural Resources Research 2018, 28, (3), 645-664.

[28] Wang, Q.; Zhang, J.; Shu, S.; Lai, C.; Xu, B.; Sun, H., Genesis of the Xiuwenghala Gold Deposit in the Beishan Orogen, Northwest China: Evidence from Geology, Fluid Inclusion, and H-O-S-Pb Isotopes. Resource Geology 2019, 69, (2), 211-226.

[29] Jiang, S.; Nie, F.; Liu, Y., Gold Deposits in Beishan Mountain, Northwestern China. Resource Geology 2004, 54, (3), 325-340.

[30] Chen, S.; Guo, Z.; Qi, J.; Zhang, Y.; Pe-Piper, G.; Piper, D. J. W., Early Permian volcano-sedimentary successions, Beishan, NW China: Peperites demonstrate an evolving rift basin. Journal of Volcanology and Geothermal Research 2016, 309, 31-44.

Point 10: Introduction: Author should discuss the problem associated with current mapping practice, and its importance along with literature (with citation).

Response 10: We appreciate the comments. We have discussed the problem associated with current mapping practice in Line 70-73, and Line 75-82. Moreover, we also cited the literature, such as [26], [27], and [20] to prove our viewpoint.

ref:

[20] Shabankareh, M.; Hezarkhani, A., Application of support vector machines for copper potential mapping in Kerman region, Iran. Journal of African Earth Sciences 2017, 128, 116-126.

[26] Fang, Z.; Wang, Y.; Peng, L.; Hong, H., A comparative study of heterogeneous ensemble-learning techniques for landslide susceptibility mapping. International Journal of Geographical Information Science 2020, 1-27.

[27] Zhang, S.; Xiao, K.; Carranza, E. J. M.; Yang, F., Maximum Entropy and Random Forest Modeling of Mineral Potential: Analysis of Gold Prospectivity in the Hezuo–Meiwu District, West Qinling Orogen, China. Natural Resources Research 2018, 28, (3), 645-664.

Point 11: Introduction: There are many articles on this subject. The author should be explicit the novelty of the current paper along with the gap area.

Response 11: Thank you for your helpful advice. We have explicated the novelty of the current paper along with the gap area in Line 62-82. In this section, we further elaborated the novelty of the current paper by reviewing the history of previous research.

Point 12: Introduction: Author should narrate the merits and demerits of prospectivity mapping methods.

Response 12: We appreciate the comments. We have added the merits and demerits of prospectivity mapping methods in Line 45-82.

Point 13: Materials and Methods: Why authors have selected Beishan region for the present research?

Response 13: Thank you for your suggestion. We have added the reason in Line 103-106.

Point 14: Conclusion: These are observations not conclusions/recommendations. This section needs to be crisp and a summary of the findings rather than discussion. Please revise it accordingly. Also, request not use bullet point in this section.

Response 14: We appreciate the comments. We have revised the conclusion section according to your suggestion. We simplified the original part into one paragraph for summarizing the findings and added two paragraphs in order to summarize the limitations and further work in the future.

Point 15: References: There are many articles on this subject. The author needs to review all of these published articles.

Response 15: We appreciate the comments. We have reeviewd all of these published articles on this subject as many as possible. And we added 18 references to this article, including [5], [6], [7], [11], [12], [18], [24], [26], [28], [29], [30], [34], [52], [54], [59], [60], [61], and [62].

ref:

[5] Li, B.; Liu, B.; Guo, K.; Li, C.; Wang, B., Application of a Maximum Entropy Model for Mineral Prospectivity Maps. Minerals 2019, 9, (9), 556.

[6] Yousefi, M.; Nykänen, V., Data-driven logistic-based weighting of geochemical and geological evidence layers in mineral prospectivity mapping. Journal of Geochemical Exploration 2016, 164, 94-106.

[7] Leite, E. P.; de Souza Filho, C. R., Probabilistic neural networks applied to mineral potential mapping for platinum group elements in the Serra Leste region, Carajás Mineral Province, Brazil. Computers & Geosciences 2009, 35, (3), 675-687.

[11] Li, X.; Yuan, F.; Zhang, M.; Jowitt, S. M.; Ord, A.; Zhou, T.; Dai, W., 3D computational simulation-based mineral prospectivity modeling for exploration for concealed Fe–Cu skarn-type mineralization within the Yueshan orefield, Anqing district, Anhui Province, China. Ore Geology Reviews 2019, 105, 1-17.

[12] Porwal, A.; Carranza, E. J. M., Introduction to the Special Issue: GIS-based mineral potential modelling and geological data analyses for mineral exploration. Ore Geology Reviews 2015, 71, 477-483.

[18] Zhang, D.; Agterberg, F.; Cheng, Q.; Zuo, R., A Comparison of Modified Fuzzy Weights of Evidence, Fuzzy Weights of Evidence, and Logistic Regression for Mapping Mineral Prospectivity. Mathematical Geosciences 2013, 46, (7), 869-885.

[24] Zhang, Y.; Li, M.; Han, S.; Ren, Q.; Shi, J., Intelligent Identification for Rock-Mineral Microscopic Images Using Ensemble Machine Learning Algorithms. Sensors (Basel) 2019, 19, (18).

[26] Fang, Z.; Wang, Y.; Peng, L.; Hong, H., A comparative study of heterogeneous ensemble-learning techniques for landslide susceptibility mapping. International Journal of Geographical Information Science 2020, 1-27.

[28] Wang, Q.; Zhang, J.; Shu, S.; Lai, C.; Xu, B.; Sun, H., Genesis of the Xiuwenghala Gold Deposit in the Beishan Orogen, Northwest China: Evidence from Geology, Fluid Inclusion, and H-O-S-Pb Isotopes. Resource Geology 2019, 69, (2), 211-226.

[29] Jiang, S.; Nie, F.; Liu, Y., Gold Deposits in Beishan Mountain, Northwestern China. Resource Geology 2004, 54, (3), 325-340.

[30] Chen, S.; Guo, Z.; Qi, J.; Zhang, Y.; Pe-Piper, G.; Piper, D. J. W., Early Permian volcano-sedimentary successions, Beishan, NW China: Peperites demonstrate an evolving rift basin. Journal of Volcanology and Geothermal Research 2016, 309, 31-44.

[34] Yue, Y.; Liou, J. G.; Graham, S. A., Tectonic correlation of Beishan and Inner Mongolia orogens and its implications for the palinspastic reconstruction of north China. Geological Society of America Memoirs 2001, 194, 101-116.

[52] Carranza, E. J. M., Objective selection of suitable unit cell size in data-driven modeling of mineral prospectivity. Computers & Geosciences 2009, 35, (10), 2032-2046.

[54] Beguería, S., Validation and Evaluation of Predictive Models in Hazard Assessment and Risk Management. Natural Hazards 2006, 37, (3), 315-329.

[59] Rokach, L., Taxonomy for characterizing ensemble methods in classification tasks: A review and annotated bibliography. Computational Statistics & Data Analysis 2009, 53, (12), 4046-4072.

[60] Zhang, M., Analysis on the causes and prospecting direction of gold deposits in North Mt. area of Gansu Province (in Chinese). World Nonferrous Metals 2019, (7), 297-298.

[61] Jiang, S.; Nie, F., Geology and ore genesis of the Nanjinshan gold deposit in Beishan Mountain area, northwestern China. Mineral Deposit Research: Meeting the Global Challenge 2005, 537-540.

[62] Liu, X.; Wang, Q., Tectonics of orogenic belts in Beishan Mts., western China and their evolution (in Chinese). Geosci. Research 1995, 28, 37-48.

Once again, thank you very much for your constructive comments and suggestions which would help us both in English and in-depth to improve the quality of the article.

Round 2

Reviewer 2 Report

Authors have incorporated the comments raised in the last review.